# Classification of Breast Cancer and Grading of Diabetic Retinopathy & Macular Edema using Ensemble of Pre-trained Convolutional Neural Networks

**Mohammed Safwan**

kp.mohd.safwan@gmail.com

**Sai Saketh Chennamsetty**

sakari1994@gmail.com

**Avinash Kori**
Department of Engineering Design
Indian Institute of Technology Madras
Chennai, India
koriavinash1@gmail.com

**Varghese Alex Kollerathu**
Department of Engineering Design
Indian Institute of Technology Madras
Chennai, India
varghesealex90@gmail.com

**Ganapathy Krishnamurthi**
Department of Engineering Design
Indian Institute of Technology Madras
Chennai, India
gankrish@iitm.ac.in

## Abstract

In this manuscript, we present a deep learning based approach for detection and classification of medical conditions such as classification of breast cancer and grading of diabetic retinopathy and macular edema. The performance of a convolutional neural network is dependent on the architecture of the network, amount of training data and data pre-processing. Transfer learning is oft utilized in deep learning so as to counter the limited availability of high quality annotated data. Hence, we create an ensemble of pre-trained classifiers by making use of models with different topologies and data normalization schemes. In general, the variance associated with an ensemble of classifiers is lower compared to a single classifier and thus generalizes better on unseen data. An F1 score based model pruning technique was utilized for deciding the optimal number of classifiers in the ensemble. The proposed technique was tested on two separate biomedical image challenges, namely the (1) classification of breast cancer from histology images [BACH-2018] and (2) grading of diabetic retinopathy and macular edema from fundus images [IDRiD-2018]. On the histology data, our technique was adjudged jointly as the top performing algorithm while for the task of diabetic retinopathy grading, the technique was declared as the $4^{th}$ best performing algorithm.

## 1 Introduction

For a variety of classification and pattern recognition based tasks, Convolutional Neural Networks (CNN) have outperformed the traditional machine learning approaches and is currently the technique

1st Conference on Medical Imaging with Deep Learning (MIDL 2018), Amsterdam, The Netherlands.

Table 1: Breast cancer histology images dataset

| Normal | Benign | In-Situ | Invasive |
|--------|--------|---------|----------|
| 100    | 100    | 100     | 100      |

Table 2: Diabetic Retinopathy and Macular Edema fundus images dataset

|                        | Grade 0 | Grade 1 | Grade 2 | Grade 3 | Grade 4 |
|------------------------|---------|---------|---------|---------|---------|
| Diabetic Retinopathy   | 134     | 20      | 136     | 74      | 49      |
| Diabetic Macular Edema | 177     | 41      | 195     |         |         |

that produces state of the art performance [14, 18, 19]. The superior performance of the CNNs comes at the cost of requiring millions of high quality labeled data for training the network. The presence of labeled data of this magnitude in the domain of medical image analysis is extremely rare. In circumstances as stated above, a transfer learning based approach can be utilized, wherein the model is first trained on a large labeled dataset such as natural images or digits and is then fine-tuned for the task of choice using the limited dataset. This approach has been widely used for numerous medical image analysis applications such as detection of thoraco-abdominal lymph node and Interstitial Lung Disease (ILD) from medical volumes [17], polyp detection from colonoscopy video frames [20], classification of mammograms [5].

The performance of CNNs is dependent on the architecture of the model and the connectivity pattern between various layers in the model. However, there exists neither a single architecture nor a connectivity pattern which guarantees best or ideal performance for a given task. It is observed that an ensemble of classifiers typically outperforms a single classifier, as an ensemble reduces the variance in the final prediction [12]. In the context of CNNs, an ensemble of classifiers can be built by either varying the architecture and the connectivity pattern of the network and/or by training the networks on data pre-processed using different variants of data normalization schemes.

This manuscript explains our approach to automate detection and diagnosis of medical conditions by making use of transfer learning and ensembling of multiple classifier models. F1 score was used as the metric to measure the performance of a classifier so as to prune the number of classifiers that constitute the ensemble. The efficacy of the proposed approach was evaluated by participating in two distinct biomedical challenges namely, (1) the classification of breast cancer from histology slides and (2) the grading of Diabetic Retinopathy (DR) and Diabetic Macular Edema (DME) from fundus images. The technique was adjudged as the top performing algorithm in breast tumor classification challenge and $4^{th}$ best entry in the challenge of diabetic retinopathy and diabetic macular edema grading.

## 2 Materials and Methods

### 2.1 Data

The breast cancer histology images were made available as part of the ICIAR Grand Challenge on BreAst Cancer Histology (BACH) images [1]. The dataset composed of 400 images and each image was classified as one of the four classes namely "Normal", "Benign", "In-Situ" and "Invasive". In the given dataset, each class comprises of equal number of images, Table 1. Further details about the dataset is given in Appendix 5.1.

The fundus images data was made available as part of the recently concluded diabetic retinopathy segmentation and grading challenge [3]. Of the 3 sub-challenges, we made use of the dataset provided for sub-challenge 2, i.e., grading of diabetic retinopathy and macular edema. The various grades of DR and DME and number of images per class are tabulated in Table 2. Further details about the data and the manifestation of each grade in the fundus image dataset is given in Appendix 5.1.

### 2.2 Convolutional Neural Networks

Traditional CNNs such as AlexNet [14], VGG-16, VGG-19 [18], comprises of convolutional layers, pooling layers and fully connected layer. Max pooling is one of the oft used pooling technique so as to

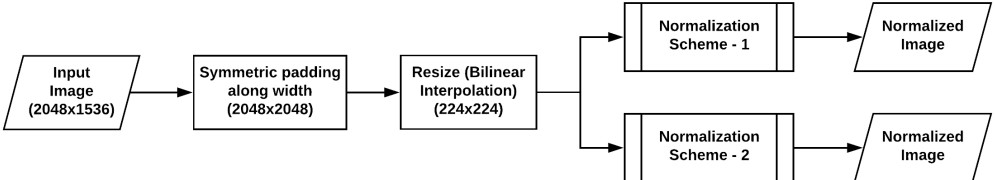

Figure 1: Pre-processing pipeline. The pre-processing pipeline comprises of padding images, resizing of image using bilinear interpolation and normalizing the image using different image statistics. The image above illustrates the pre-processing pipeline used for the breast histology images

reduce the spatial dimension of the feature maps as the depth of the network increases. Furthermore, it also aids in learning translation invariant features from the data. As the depth of the network increases, the features learned by the network becomes more abstract and thus the features learned by a layer are typically considered to be a higher level representation of the data than the features learned in the preceding layers in the network. The features of the highest abstraction are then vectorized to form the fully connected layers. In traditional CNNs, the class associated to an incoming data is predicted based on the activation produced by the penultimate layer in the network. The downside of using traditional CNNs are (1) the presence of multiple fully connected layers which lead to a drastic increase in the number of trainable parameters, and (2) making decision or the classification based solely on the highest level of abstraction of the data.

Introduction of skip and residual connections between different layers of the network aid in building deeper and newer variants of CNNs such as deep Residual Networks (ResNets) [9] and Densely connected Networks (DenseNets) [10]. When compared to traditional networks, both ResNets and DenseNets comprise of fewer number of fully connected layers, resulting in fewer number of trainable parameters. Despite having fewer parameters, the presence of skip and residual connections in the Residual Networks and Densely connected Networks help in achieving better classification performance as the decision making is based on both high and low level features extracted by the network.

We make use of 4 variants of densely connected CNNs and 5 variants of deep residual CNNs pre-trained on natural images (ImageNet dataset) [6] for all tasks. The pre-trained networks were made available by PyTorch [16].

### 2.3  Pre-processing of Data

Pre-processing of data comprises of 2 steps, namely:

- Resizing of images
- Normalization of images

Images from both the datasets were padded appropriately to attain similar dimension along the height and width of the image. The images were then resized to a dimension of $224{\times}224$ and then normalized to have zero mean and unit variance using Eq. 1. In Equation 1, $\mu$ and $\sigma$ denote the statistics (mean and standard deviation) of the dataset. The images could be normalized using either:

- $Normalization\,scheme-1$: Normalize the images using image statistics (mean and standard deviation) of the data that was used to pre-train the models [ImageNet].
- $Normalization\,scheme-2$: Normalize images using image statistics of the data that is relevant to the task at hand, i.e., image statistics from the training data of histology image or fundus images.

$$Normalized\,Image = \frac{Image - \mu}{\sigma} \tag{1}$$

The pipeline for pre-processing the data is given in Fig. 1.

## 2.4 Training of Networks

ResNet-18, ResNet-34, ResNet-50, ResNet-101, ResNet-152, DenseNet-121, DenseNet-169, DenseNet-201 and DenseNet-161 were the models used for all tasks. Each model was trained independently on the data normalized using $Normalization\ scheme-1$ and $Normalization\ scheme-2$. The number of neurons in the decision layer of the network was set appropriately based on the number of classes associated with the task. The models were initialized with the pre-trained weights and the parameters of network were fine-tuned based on the cross entropy loss. Each model was trained for 50 epochs with Adam [13] as the optimizer with a learning rate of 0.0001. Further details related to training networks for each task is given in Appendix 5.2.

## 2.5 Model Pruning

During inference, feeding the data to each of the trained models in the ensemble to generate final prediction is a computational overhead as the ensemble comprises of a total of 18 classifiers (9 per normalization scheme). More importantly using all models in the ensemble does not guarantee a good performance. In literature, pruning of classifiers is the technique used to reduce the complexity of the ensemble so as to retain the high performing classifiers and thereby achieve good performance. We use F1 score as the measure for calculating the classification capability of each model in the ensemble.

F1 score is given by Eq. 2, wherein TP, FP and FN represents the number of true positives, false positives and false negatives generated by a classifier. F1 score is also defined as the harmonic mean of precision and recall (sensitivity) and is often used to compute the performance of classifiers in scenarios where class imbalance amongst classes exists in the data. In the context of bio-medical applications, F1 score seems to be the most natural choice of metric to be used as the data imbalance amongst various classes is acute when compared to classes arising from natural images or digits.

$$F1 = \frac{2 \times TP}{2 \times TP + FP + FN} \tag{2}$$

From the ensemble, the model yielding highest F1 score ($F1_{max}$) and the models with F1 score at least 95% of $F1_{max}$ were retained while the rest were discarded. This procedure aids in reducing the number of models in ensemble from 18 to 3-6 models.

## 2.6 Testing

The inference associated to each task differs marginally, however the overall inference pipeline comprises of pre-processing the test image using different normalization schemes and feeding the normalized images to the appropriate models, i.e., models retained after pruning, in the ensemble. For a test data, each model in the ensemble predicts a class leading to an array of predictions wherein the length of array is equivalent to the number of models in the ensemble. From the array of predictions, a majority voting scheme was used to produce the final class associated with the image. More details regarding testing of models for each task is given in Appendix 5.3.

## 3 Results and Discussion

The models trained for both breast cancer classification and grading of diabetic retinoapthy & macular edema were tested on both held out test data and challenge data provided by the respective organizers.

### 3.1 Performance of breast cancer classifier on held out test data

We observed that only 3 models, i.e., DenseNet-161 and ResNet-101 trained on data prepared using $Normalization\ Scheme-1$ and a DenseNet-161 trained on data normalized with $Normalization\ Scheme-2$ satisfied the model pruning criteria as explained in Section 2.5. These models were tested on a held out test dataset ($n=40$) with equal number of images from each class.

Table 3 (a-c) shows the performance of each of the model in the ensemble. We observe that ensembling the predictions made by each of the model reduces the number of mis-classifications and thereby achieves an accuracy of 97.5%, Table 3 (d). Further performance metrics such as sensitivity,

Table 3: Confusion matrix showing result achieved by different models

(a) Densenet-161-NS-2

| | | Prediction | | | |
|---|---|---|---|---|---|
| | | Benign | Insitu | Invasive | Normal |
| Truth | Benign | 8 | 0 | 2 | 0 |
| | Insitu | 1 | 9 | 0 | 0 |
| | Invasive | 3 | 0 | 7 | 0 |
| | Normal | 1 | 0 | 0 | 9 |

(b) Resnet-101-NS-1

| | | Prediction | | | |
|---|---|---|---|---|---|
| | | Benign | Insitu | Invasive | Normal |
| Truth | Benign | 9 | 0 | 1 | 0 |
| | Insitu | 0 | 9 | 0 | 1 |
| | Invasive | 1 | 0 | 9 | 0 |
| | Normal | 0 | 0 | 0 | 10 |

(c) Densenet-161-NS-1

| | | Prediction | | | |
|---|---|---|---|---|---|
| | | Benign | Insitu | Invasive | Normal |
| Truth | Benign | 8 | 0 | 1 | 1 |
| | Insitu | 0 | 10 | 0 | 0 |
| | Invasive | 0 | 0 | 10 | 0 |
| | Normal | 0 | 0 | 0 | 10 |

(d) Ensemble

| | | Prediction | | | |
|---|---|---|---|---|---|
| | | Benign | Insitu | Invasive | Normal |
| Truth | Benign | 9 | 0 | 1 | 0 |
| | Insitu | 0 | 10 | 0 | 0 |
| | Invasive | 0 | 0 | 10 | 0 |
| | Normal | 0 | 0 | 0 | 10 |

Table 4: Performance of the proposed technique on the breast cancer classification challenge data

| Position | Team Number | Accuracy (%) |
|---|---|---|
| 1 | 216 (Ours) | 87 |
| 1 | 248 | 87 |
| 3 | 1 | 86 |

specificity and the ability of the network to distinguish between aggressive and less malignant tissues are given in Appendix 5.4. The performance of using traditional networks such as AlexNet and the boost in performance upon using pre-trained weights are given in Appendix 5.5.

### 3.2 Performance of breast cancer classifier on challenge data

The efficacy of the proposed technique was evaluated by computing performance of the technique on the test data provided by the challenge organizers. A total of 100 images without their associated ground truths were provided to all the participating teams in the challenge and performance of each team was computed by the challenge organizers. On the leader-board generated by the organizers [2], the proposed algorithm shared the top spot for the best performing algorithm, Table 4.

### 3.3 Performance of Diabetic Retinopathy classifier on the held out test data

The diabetic retinopathy classifier comprises of a primary classifier ensemble and an expert classifier ensemble. Both the ensembles made up of a number of different models and the model pruning strategy reduces the number of models in the primary and expert classifiers to 4 models each. The pruned primary classifier was composed of DenseNet-121, DenseNet-201, ResNet-18 and ResNet-34, which were trained on data normalized using ImageNet statistics. The pruned expert model comprises of 3 models viz; DenseNet-169, ResNet-18 and ResNet-34 trained with data normalized using IDRiD statistics ($Normalization\ scheme - 2$) and a ResNet-18 model trained with data normalized using ImageNet statistics.

On the test data ($n = 56$), the diabetic retinopathy classifier achieved an accuracy of 85.7%, Table 5. Similar to the findings found on the breast cancer data, ensembling the predictions from a subset of classifiers provided better performance when compared to using a single classifier or all classifiers in the ensemble. Comparison in terms of classification performance upon using ensemble of classifiers, pruning of models in ensemble. are given in Appendix 5.6.

### 3.4 Performance of Diabetic Macular Edema classifier on held out test data

The Diabetic Macular Edema (DME) classifier, comprises of 2 expert classifiers, one trained to accurately classify images as either "Grade 0" DME or not, while the other expert classifier was

Table 5: Grading of DR - Confusion matrix on the held out test data. Accuracy = 85.7%

| | | Prediction | | | | |
|---|---|---|---|---|---|---|
| | | Normal | Mild | Moderate | Severe | PDR |
| Truth | Normal | 14 | 0 | 1 | 0 | 0 |
| | Mild | 2 | 10 | 0 | 0 | 0 |
| | Moderate | 1 | 0 | 11 | 2 | 0 |
| | Severe | 0 | 0 | 1 | 8 | 0 |
| | PDR | 0 | 0 | 1 | 0 | 5 |

Table 6: Grading of DME - Confusion matrix on the held out test data. Accuracy = 95.45%

| | | Prediction | | |
|---|---|---|---|---|
| | | Grade 0 | Grade 1 | Grade 2 |
| Truth | Grade 0 | 18 | 1 | 0 |
| | Grade 1 | 0 | 5 | 0 |
| | Grade 2 | 0 | 1 | 19 |

trained to classify images as "Grade 2" DME or not. After model pruning, each expert classifier ensemble comprised of 3 models; DenseNet-161, DenseNet-169, Densenet-201 for "Grade 0" expert and DenseNet-161, ResNet-34 and ResNet-50 for "Grade 2" expert. On the held out test data ($n = 44$), the trained diabetic macular edema classifier achieved an accuracy of 95.45%, Table 6.

### 3.5 Joint performance of Diabetic Retinopathy & Macular Edema classifier on the entire IDRiD training dataset

On the entire IDRiD training data comprising of 413 images, performance of the Diabetic Retinopathy classifier and Diabetic Macular Edema classifier are tabulated in Table 7 (a,b). For their respective tasks, the Diabetic Retinoapthy classifier and the Diabetic Macular Edema classifier achieved an accuracy of 88.80% and 96.85% respectively, Table 7 (c).

The objective of the IDRiD challenge (sub-challenge 2) was to predict the grade of diabetic retinopathy as well as the grade of macular edema in a fundus image. On the entire training data provided by the organizers ($n = 413$), the diabetic retinopathy classifier and macular edema classifier jointly achieved an accuracy of 86.68%. For this task, an image is considered as a true positive if and only if the classifiers predict both the grade of retinopathy and macular edema appropriately in the image. Due to the reason stated above, achieving good performance for the task of jointly predicting the grade of retinopathy and macular edema given an image was more challenging when compared to attaining a good performance on individual tasks. On the entire training data, we observe that the performance of classifiers for the joint task is slightly lower when compared to the performance of the classifiers for the individual tasks, Table 7 (c).

### 3.6 Performance of Diabetic Retinopathy and Diabetic Macular Edema classifier on the challenge data

The performance of the classifiers was tested by participating in the recently concluded IDRiD challenge. The performance of all participating teams were evaluated by providing an on-site challenge data comprising of 103 images. On the challenge data, our approach shared the spot of $4^{th}$ best performing algorithm, Table 8. Considering grading of diabetic retinopathy and macular edema as separate challenges or tasks, the performance of each participating team for the individual task will be made available by the organizers in their post-conference journal.

The models of the top 3 performing teams were pre-trained on the Kaggle diabetic retinopathy challenge dataset [4]. The aforementioned database comprises of 35,126 fundus images arising from 5 classes similar to the diabetic retinopathy classes in IDRiD. From the results, we observed that, using limited labeled data, i.e., 1% of data used by the top 3 teams, transfer learning approaches along with good model pruning scheme could yield comparable and competitive performance.

Table 7: Performance of Diabetic Retinopathy (DR) classifier and Diabetic Macular Edema (DME) classifier on training data ($n = 413$)

(a) DR classifier

| | | Prediction | | | | |
|---|---|---|---|---|---|---|
| | | Normal | Mild | Moderate | Severe | PDR |
| Truth | Normal | 130 | 0 | 4 | 0 | 0 |
| | Mild | 7 | 12 | 1 | 0 | 0 |
| | Moderate | 6 | 1 | 127 | 2 | 0 |
| | Severe | 0 | 0 | 11 | 63 | 0 |
| | PDR | 0 | 0 | 6 | 8 | 35 |

(b) DME classifier

| | | Prediction | | |
|---|---|---|---|---|
| | | Grade 0 | Grade 1 | Grade 2 |
| Truth | Grade 0 | 172 | 5 | 0 |
| | Grade 1 | 2 | 38 | 1 |
| | Grade 2 | 2 | 3 | 190 |

(c) Accuracy of classifiers for individual and joint task

| Task | Network | Accuracy (%) |
|---|---|---|
| Grading of Diabetic Retinopathy | DR classifier | 88.80 |
| Grading of Diabetic Macular Edema | DME classifier | 96.85 |
| Grading of Diabetic Retinopathy + Diabetic Macular Edema | DR classifier & DME classifier | 86.68 |

Table 8: Performance on the IDRID on-site challenge data ($n = 103$).

| Position | Team Name | Accuracy (%) |
|---|---|---|
| 1 | lzyuncc | 63.11 |
| 2 | VRT | 55.34 |
| 3 | Mammoth | 51.46 |
| 4 | AVSASVA (Ours) | 47.57 |
| 4 | HarangiM1 | 47.57 |
| 6 | HarangiM2 | 40.78 |

## 4 Conclusion

We present a transfer learning and model ensembling technique for two distinct medical image classification related tasks. Limited amount of high quality labeled training data was circumvented by initializing the networks with weights pre-trained on the natural images. Each model in the ensemble was trained independently on data normalized using two different schemes. From the assemblage of models in the ensemble, a pruning technique based on F1 score was used as the metric to prune the number of networks in the ensemble. This technique was found to

- Reduce the number of models in the ensemble, depending on the task, from 18 to 3-6.
- Provide better performance when compared to using a single classifier or all the models in the ensemble.

The efficacy of the networks trained using the scheme proposed in the manuscript were tested by participating in the BACH-2018 and IDRiD challenges. On the challenge data ($n = 100$), for the task of classifying grade of breast cancer from histology images, our technique shared the spot of the best performing algorithm. For the task predicting grade of diabetic retinopathy and diabetic macular edema from fundus image, on the challenge data ($n = 103$), our submission shared the $4^{th}$ position on the leader-board. Based on the results attained on the challenge data from both challenges, we conclude that:

- For a variety of medical image classification related tasks where the amount of labeled data is sparse, transfer learning is a better choice when compared to training network from scratch.
- Ensembling predictions from multiple models aids in reducing the variance and thereby improves the overall performance.
- Compared to models trained on extremely large dataset, our technique produce comparable and good performance on the unseen data.

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

## 5   Appendix

### 5.1   Data

We made use of dataset from two recently concluded medical image classification challenges namely, the "ICIAR-2018 Grand challenge on BreAst Cancer Histology images" [BACH-2018] and "Diabetic Retinopathy Segmentation and Grading Challenge" [IDRiD].

The breast cancer histology image dataset was composed of 400 high-resolution (2040×1536 pixels), uncompressed, and annotated Hematoxylin and Eosin (H&E) stained images. According to the predominant cancer type, each image was labeled with one of four classes, namely normal tissue, benign lesion, in situ carcinoma and invasive carcinoma. The images were normalized using [15] in order to eliminate the inconsistencies in the staining process during slide preparation. In the given training dataset, each class comprises of equal number of images. Fig. (2 (a-d)) illustrates the images from four classes when viewed under a microscope after appropriate staining and normalization.

The fundus images used for DR and DME grading were made available as part of the Segmentation and Grading challenge held in conjunction with the 2018 IEEE International Symposium on Biomedical Imaging (ISBI). The images where acquired using a Kowa VX-10 alpha digital fundus camera with a resolution of 4288×2848 pixels. Based on international standards of clinical relevance, each image was complemented with the information regarding the disease severity level of diabetic retinopathy, and diabetic macular edema. Fig. (3 (a-e)) & (3 (f-h)) illustrates different grades and Tables 9 & 10 lists the criteria used for grading diabetic retinopathy and diabetic macular edema respectively. The limited number of "Mild" NPDR class in the IDRiD dataset was addressed by using a publicly available dataset [11].

### 5.2   Training of Networks

For each task, several variants of Residual Networks (ResNets) and Densely connected convolutional Networks (DenseNets) form the ensemble. Each model in the ensemble was trained independently on data pre-processed using different normalization schemes.

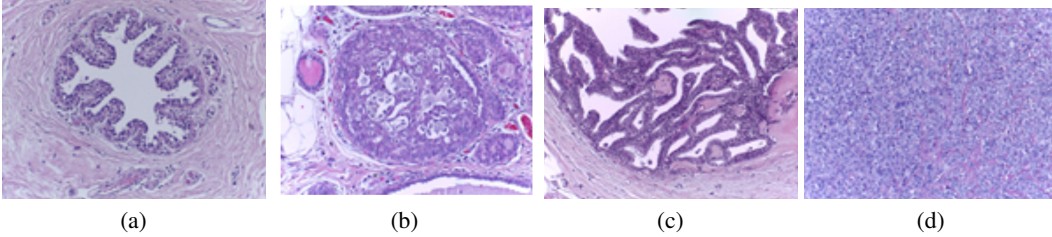

| (a) | (b) | (c) | (d) |

Figure 2: Breast histology images. (a) Normal tissue (b) Benign lesion (c) in situ carcinoma (d) Invasive carcinoma

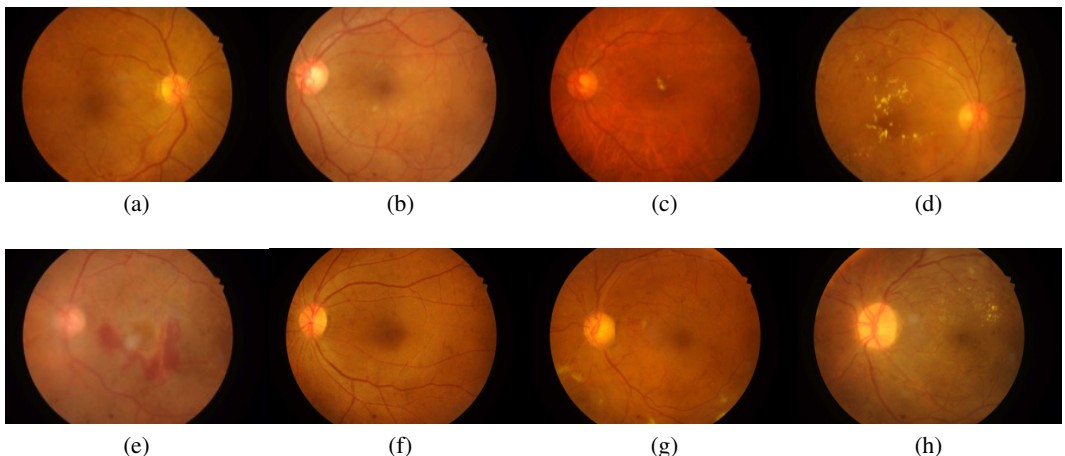

Figure 3: Grades of Diabetic Retinopathy (a-e) and grades of Diabetic Macular Edema (f-h). (a) Normal (b) Mild NPDR (c) Moderate NPDR (d) Severe NPDR (e) PDR (f) Grade 0 DME (g) Grade 1 DME (h) Grade 2 DME

Table 9: Diabetic Retinopathy severity scale

| Grade | Criteria |
|---|---|
| Grade 0: No apparent retinopathy | No visible sign of abnormalities |
| Grade 1: Mild – NPDR | Presence of microaneurysms only |
| Grade 2: Moderate – NPDR | More than just microaneurysms but less than severe NPDR |
| Grade 3: Severe – NPDR | Any of the following:
• > 20 intraretinal hemorrhages
• Venous bleading
• Intraretinal microvascular abnormalities
• no signs of PDR |
| Grade 4: PDR | Either or both of the following:
• Neovascularization
• Vitreous/pre-retinal hemorrhage |

Table 10: Diabetic Macular Edema severity scale

| Grade | Criteria |
|---|---|
| Grade 0 | No Apparent hard exudate(s) |
| Grade 1 | Presence of hard exudate(s) outside the radius of one disc diameter from the macula center |
| Grade 2 | Presence of hard exudate(s) within the radius of one disc diameter from the macula center |

### 5.2.1 Training of networks for breast cancer classification

Each network pre-trained on ImageNet database was fine-tuned with 70 images from each class while the networks were validated and tested on 20 and 10 images from each class respectively. The decision layer of the network was modified from 1000 neurons to 4 neurons (Normal, In-situ, Benign and Invasive). The parameters of the network were learned by minimizing the cross entropy between predictions made by the network and the true label/class associated with the data. The networks were trained for 50 epochs with the learning rate set to 0.0001 and ADAM as the optimizer.

### 5.2.2 Training of networks for grading of Diabetic Retinopathy

For the task of automatic grading of diabetic retinopathy, two sets of ensemble of CNNs viz; "primary classifier" and "expert classifier" were used. The primary classifier was trained to classify a fundus image as one of the 4 classes, namely "Normal", "Mild NPDR", "Moderate NPDR" or "S-(N)-PDR" where S-(N)-PDR was a newly created class by clubbing "Severe NPDR" and "PDR". Each model in the ensemble was trained and validated using 70% and 20% of the entire training data ($n = 502$). All networks were initialized with pre-trained weights and all other hyper-parameters related to training of networks such as learning rate, choice of optimizer were similar to the ones used for training the breast cancer classification.

The expert classifier was trained to demarcate fundus images as either "Severe NPDR" or "PDR". The models in the expert classifiers were trained and validated exclusively on the images with the aforementioned classes. Apart from difference in the number of classes, the training regime and all hyper-parameters such as learning rate, optimizer were similar to the ones used to train the models in the primary classifier.

### 5.2.3 Training of networks for grading of Diabetic Macular Edema

The task of DME grading was to classify each fundus image into one of the three grades depending on the criteria given in Table 10. Two sets of ensemble of networks, namely "Expert Class 0" model and "Expert Class 2" model were trained in a one versus rest approach for the same. "Expert Class 0", an ensemble of CNNs was trained to distinguish between the classes "No apparent exudates" (Grade 0) and " Presence of exudates" (Grade 1 & Grade 2). On the other hand, "Expert Class 2" was trained to classify fundus images as either "Grade 2" DME or not. Training the ensemble in the proposed one vs rest fashion aids in mitigating the acute class imbalance in the given dataset.

## 5.3 Testing of Networks

### 5.3.1 Testing of networks trained for breast cancer classification

The histology images to be tested were resized and normalized appropriately by passing the images through the pre-processing pipeline. For a normalized test data, each model in the pruned ensemble predicts severity of the lesion in the image. On the predictions made by different models in the pruned ensemble, a majority voting criterion was utilized to assign the final prediction. The testing regime is illustrated in Fig. 4.

### 5.3.2 Testing of networks trained for grading of Diabetic Retinopathy

For the task of automated grading of diabetic retinopathy, we observed test time augmentation had a positive impact on the overall performance of technique. In this work, the test time augmentation comprises of resizing each fundus image to have a fixed dimension and then extracting 10 cropped versions of the resized image. The first 5 cropped versions of the image were acquired by cropping the resized image (256×256) along the 4 corners and the center of the image to the required dimension (224×224). The following 5 crops were attained by flipping the image horizontally and repeating the cropping procedure stated above.

The grading of diabetic retinopathy from fundus images were obtained by making use of a "primary" and an "expert" classifier. The primary classifier, an ensemble of CNNs, was trained to classify the input as one of 4 classes, namely "Normal", "Mild NPDR", "Moderate NPDR" or "S-(N)-PDR". The expert classifier, an ensemble of networks, was trained to distinguish between "Severe NPDR" or "PDR".

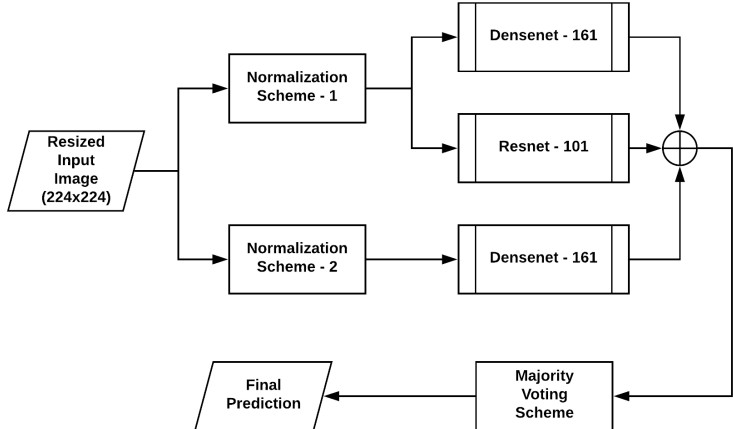

Figure 4: Proposed testing pipeline for classification of breast cancer from histology images.

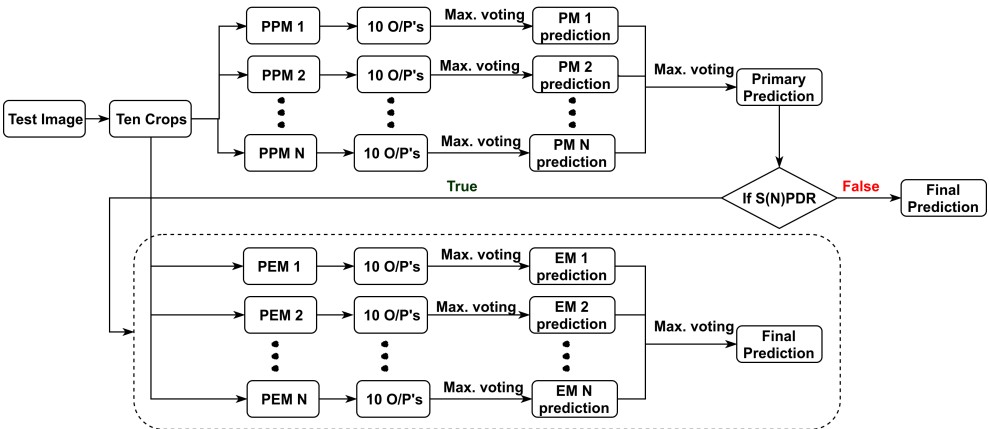

Figure 5: Inference pipeline for automated diabetic retinopathy grading. In the image, PPM: Pruned Primary Model, PM $X$: prediction made by Primary Model $x$, PEM: Pruned Expert Model, EM $X$: prediction made by Expert Model $x$, O/P's: outputs

Each of the cropped images was then normalized appropriately and fed to the models constituting the primary classifier. For each of the 10 different cropped versions of the image, a model in the primary classifier predicts the degree of diabetic retinopathy in it. For a network ($N$) in the primary classifier, class assigned to a test image was attained by employing a max voting scheme on the predictions made on the various cropped versions of the image by $N$. Thus every network in the primary classifier assigns a class to the test image.

From the array of predictions made by different networks in the primary classifier, a second max voting scheme was utilized to get the primary classification. For a given test image, if the primary prediction was "S-(N)-PDR", then the 10 cropped versions of the image were fed to each network constituting the "Expert Classifier". For a network $E$ in the expert classifier, the procedure related to assigning a class based on the 10 crops was similar to the one used in the primary classifiers. Based on the prediction made by individual networks in the expert classifier, a second max voting scheme was used to assign the final prediction to the provided test image. Figure 5 illustrates the aforementioned testing scheme for automated grading of diabetic retinopathy.

### 5.3.3 Testing of networks trained for grading of Diabetic Macular Edema

In our proposed approach for grading of diabetic macular edema, test images are fed to both "Expert Class 0" (Model 1) and "Expert Class 2" (Model 2) during inference. Model 1 predicts the

Table 11: DME Decision Maker

|  | Model 1 (Absence of hard exudates) | Model 2 (Presence of Grade 2 exudates) | Final Prediction |
|---|---|---|---|
| Case 1 | True | False | Grade 0 |
| Case 2 | False | True | Grade 2 |
| Case 3 | False | False | Grade 1 |
| Case 4 | True | True | Grade 2 |

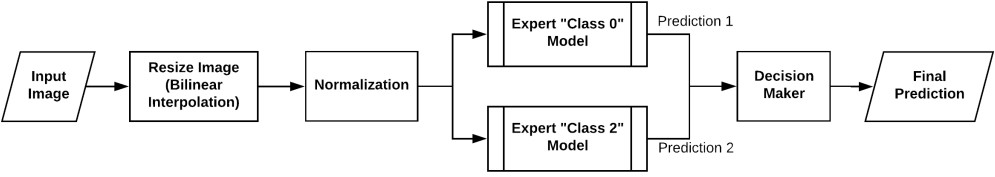

Figure 6: Testing pipeline for Automated Diabetic Macular Edema grading.

presence/absence of exudates whereas Model 2 predicts the presence of grade 2 exudates. Predictions from both the networks were sent to a decision maker which returns the final prediction for the test image depending on the logic given in Table (11). Note that the image was classified as grade 2 DME whenever Model 2 prediction says so, even if Model 1 predicts that hard exudates are absent. This rule was added to the logic to improve the sensitivity associated to "Grade 2" DME. Fig. 6 illustrates the complete testing pipeline used for automated grading of DME.

### 5.4 Performance of the proposed technique for classification of breast cancer from histology images

On the held out test data ($n = 40$), it was observed that the ensemble of networks produced better performance when compared to individual models. For different classes in the dataset, Table 12 tabulates the performance of the proposed model in terms of sensitivity, specificity and precision.

Clinically, the breast cancer types normal and benign are less malignant, and are considered as "Non-Carcinoma", while the types in-situ and invasive are relatively more aggressive and hence is collectively termed as "Carcinoma". The performance of network in distinguishing more aggressive cancer from less malignant cancer is given in Table 13. For this 2-class problem, the proposed scheme achieved a sensitivity, specificity and precision of 95%, 100% and 95% respectively.

### 5.5 Effect of using traditional networks and initializing the networks with pre-trained weights

Since the models were trained on a large cohort of natural images (ImageNet), kernels in the networks would respond to edges, patterns and textures in the images. Thus initializing the network with pre-trained weights is analogous to providing a good weight initialization scheme to the network. To evaluate the effect of using pre-trained networks, we trained a traditional network (AlexNet) from scratch to compare the results to that of a pre-trained network.

Table 12: Performance metric of the proposed network on the held out test data.

|  | Sensitivity (%) | Specificity (%) | Precision (%) |
|---|---|---|---|
| Normal | 100 | 100 | 100 |
| Benign | 90 | 100 | 90 |
| In-Situ | 100 | 100 | 100 |
| Invasive | 100 | 96 | 90 |

Table 13: Performance of network in distinguishing more aggressive tissue from less malignant tissue

|  | | Prediction | |
|---|---|---|---|
|  | | Carcinoma | Non-Carcinoma |
| Truth | Carcinoma | 20 | 0 |
|  | Non-Carcinoma | 1 | 19 |

Table 14: Performance of different networks. Except for DenseNet-161(pre-trained), weights and biases in other networks were initialized using Xavier initialization.

| Model | Validation Accuracy | Testing Accuracy | No. of Parameters |
|---|---|---|---|
| AlexNet | 0.68 | 0.53 | 57020228 |
| ResNet-18 | 0.71 | 0.63 | 11178564 |
| DenseNet-121 | 0.79 | 0.65 | 6957956 |
| DenseNet-161 | 0.79 | 0.80 | 26480836 |
| DenseNet-161 (pre-trained) | 0.98 | 0.95 | 26480836 |

A total of four models, namely AlexNet, ResNet-18, DenseNet-121 and DenseNet-161 were trained from scratch and the weights of these networks were initialized using Xavier initialization. The performance of the above classifiers along with a DenseNet-161 initialized with pre-trained weights on the validation and held out test data is given in Table 14. It was observed that initializing the network with pre-trained weights and thus performing a transfer learning approach aided in achieving better validation and testing accuracy when compared to the models trained from scratch.

## 5.6 Effect of ensembling, pruning and test time augmentation in grading of diabetic retinopathy

The diabetic retinopathy classifier comprises of a primary classifier and an expert classifier. Both the primary and expert classifiers are ensembles of 16 convolutional neural networks each. On the held out test data ($n = 56$), individual models in the ensemble achieved an accuracy in the range of 70-74%. Upon ensembling the predictions made by all models, an improvement of 1% was observed in the primary and expert classifiers when compared to the best model in the ensemble.

The model pruning technique used in this work helps in reducing the number of models in the primary and expert classifier to 4 networks each. These networks further improve the performance by 2.7%. Test time augmentation is a technique which has been used extensively so as to reduce variance in prediction and improve the overall performance [7, 8]. In this work we make use of the 10 crop technique, wherein the image is cropped to a fixed dimension along the 4 corners and centre of the image, and flipping the image horizontally and repeating the cropping procedure. This test time augmentation along with model pruning yield an accuracy of 85.7% on the held out test data. The effect of having each of the aforementioned steps during inference is tabulated in Table 15.

Table 15: Effects of each step on the performance of the classifier.

| Method | Accuracy(%) |
|---|---|
| Without model Ensemble | 70-74(individual models) |
| With model Ensemble | 75 |
| Without model Pruning | 75 |
| With model Pruning | 76.78 |
| Without Ten Crops | 76.78 |
| With Ten Crops | 85.7 |
| Our method (Ten crops + model pruning) | 85.7 |

