# OpenReview forum: "Classification of Breast Cancer and Grading of Diabetic Retinopathy & Macular Edema using Ensemble of Pre-trained Convolutional Neural Networks"
_MIDL.amsterdam/2018/Conference — Submitted to MIDL 2018_

### Review · AnonReviewer3 · 2018-05-04
**The contribution of the paper is not clear**

**Rating:** 1
**Confidence:** 2

**Review:**

Pros:
+evaluation on biomedical chalenges
Cons:
-the contribution of the paper is not clear
-the presentation of the paper should be improved

This paper applies an ensemble of ResNets and DenseNets to two problems: 1) breast cancer classification based on histology images and 2) funds image grading. The ensemble is evaluated on biomedical challenges with good results.

I like that the paper is evaluated on biomedical image challenges. My main concern is with respect to the contribution of the paper. The method is based on standard image classification architectures and it is well known in the community that the ensembles of the models works well. Thus, the methodological contribution is rather limited.

Also, it is not clear why the ensemble presented in the paper would give the best results when compared to other ways of building model ensembles. An alternative way of having an ensemble would be to train the same model on different data folds (e. g. 10fold cross-validation). Would the approach presented in the paper be better than cross-validation based model ensemble?

The papers with limited contribution that are well executed are of value to the community. However, the presentation of the paper makes it difficult to draw conclusions. What new things do we learn from the presented approach? One suggestion on how to improve presentations might be to discuss the differences of proposed method w.r.t. the alternative approaches presented in Table 4 and Table 8. What parts of the used approach might be useful for community when tackling future biomedical challenges?

The authors claim that "transfer learning is a better choice when compared to training network from scratch"; however, I couldn't find any experiment that would compare transfer learning to training from scratch in the main body of the paper. The results supporting this claim are in Appendix and should be presented in the main body of the paper.

In general, the presentation of the paper could be improved. For example, Tables 3, 5, 6, 7(a) and 7(b) could be moved to Appendix while Tables 14 and 15 should appear in the main body of the paper.

Is model pruning criteria (95% of F1 score) computed on validation set?

The authors use oft in same parts of the paper, it should be corrected to often.

**Special Issue:**

No

---

### Review · AnonReviewer1 · 2018-05-09
**The paper is suitable as a challenge submission, but there is no additional contribution.**

**Rating:** 2
**Confidence:** 2

**Review:**

The paper proposes a framework for classification in 2D medical images exploiting 1) an ensemble of deep architectures; and 2) transfer learning, using networks pretrained on large databases of natural images. The framework is applied to the detection of breast cancer from histology images, and the grading of diabetic retinopathy -DR- and macular edema -DME- from fundus images.

The strength of the paper is to report the methodology and results for the authors' entries to the BACH 2018 (top performing algorithm) and IDRiD 2018 (4th best entry) challenges. However, as is, the paper is somewhat redundant with the challenge submissions/workshops since:
- The validation does not make a case for the proposed approach, beyond reporting challenge results. Which ideas should the community take home and why?
- The methodology itself is not novel. Both this type of pretraining, and ensembling of deep architectures, have been previously proposed (and indeed cited in the manuscript).

A potential contribution (compared for instance to [12]) could be in the model selection step, but the validation does not make a convincing case for it yet.

Minor comments & questions:

- For the DR/DME application, the drop in performance on the on-site challenge data is severe, from 86% (hold-out accuracy) down to 48%. It would be valuable to give some insight in the paper.

- For DR: Table 15 suggests that the most performance-enhancing step is the test-time augmentation via multiple crops (+9%), as opposed to ensembling (+1%) or model selection (+1-2%). This is unexpected as the paper emphasises the ensembling and model selection. The pretraining does appear to have a significant effect on the application to breast cancer detection. Can the authors comment on this?

- The weighting of classes in the F1 score (Eq. 2) should be disambiguated for the multi-class case.

- What does the validation accuracy precisely refer to (e.g., Table 14 or paragraph 5.2.2.), in contrast to test accuracy? Which parameters are tuned during the validation? Which subset of the data is used to compute the F1 score for model selection?

**Special Issue:**

No

---

### Review · AnonReviewer2 · 2018-05-09
**Classification of Breast Cancer and Grading of Diabetic Retinopathy & Macular Edema using Ensemble of Pre-trained Convolutional Neural Networks**

**Rating:** 2
**Confidence:** 3

**Review:**

The paper presents an ensemble of pre-trained convolutional neural networks to tackle the problem of image classification in the BACH challenge (histopathology image classification) and the IDRiD challenge (retina image classification).
Four DenseNet models and five ResNet models pre-trained on ImageNet data are combined, each model trained two times, using different data normalization strategy.
In total, 18 convolutional networks are trained.
At test time, all models are applied, and majority voting is considered to obtain the final prediction.

The presented solution achieved the best results in the BACH challenge (Task A).

The main contributions of this paper are:
(1) pre-trained networks are a better choice compared to training network from scratch, when limited training data is available;
(2) ensemble of models reduces the variance and therefore improves overall performance.
Although these messages are valid and supported by experimental results, none of these statements is novel in the (deep) machine learning and medical image analysis community, which limits the novelty of the contribution of this paper.

The paper contains several typos, and some details are missing or not clearly explained.
Padding is used, but the padding technique is not used.
The pruning technique is described as based on F1-score, which reduced CNNs from 18 to 3-6. It is not clear whether the final CNNs are then 3 or 6.
An accuracy of 97.5% is reported when an ensemble of networks is used, but it is not clear from the text whether this was achieved with all models or only with the ones after pruning.
In Table 3, confusion matrices for the test set (n=40) are reported, but it is not reported then how many images were used for training and validation of each network, if 40 images were held out.

**Special Issue:**

No

---

### Decision · Program_Chairs · 2018-05-15
**Paper110 Acceptance Decision**

Reject